# Treatment of Tendon Injuries in the Servicemember Population across the Spectrum of Pathology: From Exosomes to Bioinductive Scaffolds

**DOI:** 10.3390/bioengineering11020158

**Published:** 2024-02-05

**Authors:** Mikalyn T. DeFoor, Daniel J. Cognetti, Tony T. Yuan, Andrew J. Sheean

**Affiliations:** 1San Antonio Military Medical Center, Fort Sam Houston, TX 78234, USA; 2Advanced Exposures Diagnostics, Interventions and Biosecurity Group, 59 Medical Wing, Lackland Air Force Base, San Antonio, TX 78236, USA; 3Center for Biotechnology (4D Bio3), Uniformed Services University of the Health Sciences, Bethesda, MD 20814, USA

**Keywords:** tendinopathy, military servicemembers, tendon tissue engineering, nonbattle musculoskeletal injuries, NBMSKI, operational medicine

## Abstract

Tendon injuries in military servicemembers are one of the most commonly treated nonbattle musculoskeletal injuries (NBMSKIs). Commonly the result of demanding physical training, repetitive loading, and frequent exposures to austere conditions, tendon injuries represent a conspicuous threat to operational readiness. Tendon healing involves a complex sequence between stages of inflammation, proliferation, and remodeling cycles, but the regenerated tissue can be biomechanically inferior to the native tendon. Chemical and mechanical signaling pathways aid tendon healing by employing growth factors, cytokines, and inflammatory responses. Exosome-based therapy, particularly using adipose-derived stem cells (ASCs), offers a prominent cell-free treatment, promoting tendon repair and altering mRNA expression. However, each of these approaches is not without limitations. Future advances in tendon tissue engineering involving magnetic stimulation and gene therapy offer non-invasive, targeted approaches for improved tissue engineering. Ongoing research aims to translate these therapies into effective clinical solutions capable of maximizing operational readiness and warfighter lethality.

## 1. Introduction

Nonbattle musculoskeletal injuries (NBMSKIs) are a main source of disability in servicemembers, attributing to over 2.4 million healthcare visits and 25 million missed duty days on an annual basis [1]. While NBMSKIs are often less severe than injuries sustained in combat, they are far more common and result in a significantly higher number of lost duty days, nondeployable servicemembers, and medical separation rates [2]. The leading NBMSKIs for service members include tendon injuries and tendinopathies, and the leading extrinsic risk factor for NBMSKI is participation in high-risk activities. Due to the significant burden on already constrained healthcare systems, continued efforts to optimize the treatment and prevention of tendon injuries among servicemembers remain a top priority for clinicians and policymakers. 

Tendons serve as a mechanical bridge, allowing the transmission of muscle strength across the bone and joint. Tendons are continuously under mechanical stress, and eccentric loads placed on a muscle under maximal stress can lead to the rupture of the surrounding tendon. A tendon bears over 10-fold an individual’s body weight during repetitive and high-impact activity, which causes a high amount of strain [3]. Managing tendon injury is clinically challenging, especially in the setting of the highly active military servicemember population. The complete pathophysiologic profile of tendinopathy and tendon injuries remains unclear, and treatment varies across the spectrum of these injuries. Given this fact, as well as the broader functional implications of these injuries, ongoing innovation is required in order to mitigate their effects on servicemembers’ operational readiness. Therefore, the purpose of this comprehensive review is to describe emerging technologies with a focus on improving the detection, prevention, and treatment of tendon injuries in tactical athletes, with a special focus on exosomes, cell-based therapies, and bioinductive scaffolds.

## 2. The Burden of Disease among Servicemembers

Tendon injuries exist across a spectrum, ranging from low-grade tendinitis to complete tears. The prevalence of tendinopathy among servicemembers is particularly noteworthy, which may be related to the demands of continuous physical training and exposure to more intense activities performed in austere environments [4]. Tendinopathy is the leading musculoskeletal overuse injury in the military population, with the overall incidence of musculoskeletal overuse injuries being 10 times higher in servicemembers compared to the general US population [5]. Overuse injuries, such as tendinopathies, comprise 80% of all injuries in active duty servicemembers [5], and three-quarters of medically nondeployable servicemembers are related to NBMSKIs [2,6]. Four percent of US military personnel are unable to deploy because of NBMSKIs [2], which serves as a clear threat to medical readiness [6]. In theater, nonbattle injuries account for 30% of all medical evacuations, and more than 85% of servicemembers do not return to theater after medical evacuation [2]. Furthermore, the military health system (MHS) expends over USD 3 billion annually to servicemembers on limited duty status, while members continue to collect their standard pay rate and benefits [5].

Incomplete tendon tears and tendinopathies are a frequent cause of diminished physical function in young, active-duty servicemembers and are more frequent in occurrence than complete tendon ruptures. On the other hand, major tendon ruptures have increased over recent decades with increased recreational sports, and several of these injuries are commonly observed among the military population [4,7]. Activities with maximal eccentric loading, such as repetitive jumping and sprinting, pose a potential risk for tendon rupture of the lower extremity, specifically when an eccentric load is accompanied by forceful concentric contraction across a tendon [8]. 

Lower extremity tendinopathy is diagnosed in US servicemembers at a rate of 2.8% on an annual basis [4], compared to a 0.05% rate of major tendon rupture [8]. Achilles tendinopathy is reported between 15 and 24% [9], with up to one-third of these patients undergoing surgical intervention after the failure of conservative therapy. Comparatively, the prevalence of Achilles tendon rupture across US servicemembers is 7.4 per 1000 persons among officers and 6.4 per 1000 persons among enlisted members. Sixty percent of patients return to duty within 1 year with good results [9]. Although a less common tendon injury, the incidence of spontaneous patellar tendon rupture is higher among servicemembers compared to the general population, at an overall incidence of 6 per 100,000 persons, with male gender, black race, and age 35 to 44 years predicting a higher risk of patella tendon rupture [7]. Overall, 75% of active duty servicemembers returned to a prior level of physical activity, while 10% were medically separated, with an overall rupture rate of 3% in US military servicemembers. While even rarer in occurrence, rupture of the pectoralis major tendon is increasingly prevalent in the military population and is most commonly attributed to bench pressing [10]. The incidence of pectoralis major tendon rupture is 60 per 100,000 persons, with risk factors including army branch, junior officer or junior enlisted rank, and age between 25 and 34 years [11].

## 3. Implications on Operational Readiness and Injury Prevention

The primary cause of NBMSKIs is a high proportion of overuse and repetitive microtrauma injuries such as running, marching, and rucking with heavy loads and lifting heavy objects. Modifiable risk factors for these injuries include alcohol consumption, smoking, extremes of body mass index (BMI), inadequate sleep, and prior NBMSKI [12].

Overuse NBMSKIs, such as tendinopathy and acute tendon rupture, are relevant targets for prevention and risk reduction strategies, given the high incidence among servicemembers and the potential for risk mitigation by standardized adaptations in physical training [13]. Regulated exercise programming has reduced the risk of injury among servicemembers. Pre-accession and pre-deployment fitness screens can be used to identify those at risk of injury or attrition. Furthermore, unit-based strength training and conditioning coaching are the primary measures recommended to reduce the risk of injury [13]. Prevention strategies rely on the timely identification and evaluation of servicemembers with tendon injuries. As secondary risk reduction measures, athletic trainers and physical therapists positioned on-site within units and primary care clinics may reduce the number of limited duty days [14]. On a larger scale, early point-of-care screening for psychological risk factors affecting readiness and responsiveness to intervention may reduce the risk of progression to long-term disability. Furthermore, standard metrics should be developed to enable commanders and clinicians to readily identify and risk stratify servicemembers with a low projected response to treatment.

The preventative effects of non-exercise therapies for NBMSKI among the military population include shock-absorbing insoles, padded polyester socks, calcium alone or combined with vitamin D supplementation, protein supplementation, and dynamic patellofemoral braces [14]. Furthermore, seven key injury prevention strategies for evidence-based load management to minimize the risk of overuse NBMSKI are outlined in Table 1 [15]. In order to maximize preventative strategies and reduce injury with associated attrition risks, strategies include meticulous preparation of training and monitoring loads, fluid adjustments of training load, and avoiding overloading patterns by the use of flexible training times [15]. Individualization and differentiation in routine physical fitness assessments and pre-deployment training are imperative to improve the overall fitness and readiness of servicemembers [5,15]. However, this can be challenging and necessitates more drill instructors and workload per instructor, as well as higher costs per servicemember [13,15].

## 4. Review of Basic Tendon Structure 

Tendons function by transmitting muscle-generated forces to bones. Type I collagen is most prevalent in tendons and makes up 65–80% of the extracellular matrix (ECM) [16]. Elastin is also embedded in a proteoglycan–water matrix, which accounts for 1–2% of the overall tendon mass. Within the tendon, collagen fibers arrange to form fibrils, which are arranged in parallel to each other and to the tendon axis (Figure 1) [17]. Tendons generally develop in three stages: collagen fibrogenesis, linear growth, and lateral growth [16,18,19]. Tenoblasts and tenocytes, elongated fibroblasts and fibrocytes, respectively, organize between collagen fibers into a complex, organized structure to form a tendon [19]. Tropocollagen forms cross-links to generate microfibril aggregates and form the overall collagen fibril. Multiple collagen fibrils aggregate to form a collagen fiber, which supplies the basic tendon unit. Collagen fibers coalesce into bunches, which are then grouped together to form a fiber bundle. Epitenon is a fine, protective, connective tissue sheath that encases the entire tendon unit. 

## 5. Principles of Tendon Healing

Tendon healing is a complex process that includes distinct stages of inflammation, proliferation, and remodeling, as illustrated in Figure 2 [20]. Illustrating the underlying pathogenesis of tendinopathy and the phases mediating tendon healing helps investigators identify targets for the development of novel strategies to enhance tendon healing and repair. An understanding of the well-delineated, predictable stages of tendon healing is crucial, as this provides insight to guide novel approaches to intervene and improve therapies for tendon tissue engineering (Table 2).

The inflammatory stage occurs first and lasts for 48 h, consisting of erythrocytes, leukocytes, endothelial chemoattractants, and platelet infiltration. Next, macrophages infiltrate the local tissue to remove necrotic debris. This is followed from day 7 to 21 with the proliferative stage, in which macrophages and tenocytes organize the synthesis of type III collagen, which is the dominant tissue type in healing tendons, although less durable than native tissue [18]. The final stage of remodeling occurs 6–8 weeks after the inciting injury, lasts for up to 12 months, and involves ECM arrangement and replacement of type III collagen with collagen type I synthesis. During the remodeling process, the maturation and arrangement of collagen fibers occur in parallel to the direction of the applied external mechanical stress.

On the cellular level, tenocytes and tenoblasts are specialized fibroblasts that co-exist [16]. During the tendon healing phase, tenoblasts are involved in tissue repair by depositing collagen fibers. During the final phase of repair, tenoblasts are transformed into tenocytes, which proliferate within the epitenon. There are several physical as well as biological risk factors leading to tendinopathy, which include older age, repetitive loading, extreme physical exercise, and oxidative stress [21,22]. Despite the remodeling process, tendons that have undergone the complete cycle of healing do not completely restore the ECM structure and biophysical properties that match those of a native tendon. Instead, the repaired tissue mimics the properties and appearance of scar tissue, which is biomechanically inferior to the native tendon [18]. This phenomenon is due to several factors including tenocytes, which demonstrate a reduction in type I collagen and subsequently reduced tensile strength [18]. Furthermore, poor vascularity in the healing tendon creates an environment with suboptimal native healing and further scar formation.

**Table 2 bioengineering-11-00158-t002:** Phases of healing after tendon injury with targets for intervention [23].

Phase	Time	Predominant Cell Types	Cytokines	Effect
Inflammatory	0 to 48 h	Neutrophils, macrophages, TSPCs	IL-β1, TNF-α, IL-6	Pro-inflammatory cytokine migration to injury site Angiogenesis promotion Exudation and fibrin leakage
Proliferative	2 days to 6 weeks	Tenocytes, macrophages, fibroblasts	IL-6, IL-8, IL-10	Fibroblast migration and proliferation Cellularity and matrix production Collage type III synthesis and deposition
Remodeling	6 weeks to 12 months	Tenocytes, macrophages, apoptotic cells	BMPs, TGF-β, IFG-1	Mature tissue formation Collaged type I synthesis, replacement of type III collagen

BMPs = bone morphogenetic proteins; IGF = insulin-like growth factor; IL = interleukin; TSPCs = tissue-specific progenitor stem cells; TGF = transforming growth factor; TNF = tumor necrosis factor.

## 6. Signaling Pathways to Stimulate Tendon Repair

Tissue tendon engineering is a complex process that involves both chemical and mechanical signaling pathways in order to regulate cellular responses and facilitate tendon healing. Chemical and mechanical signaling pathways offer a synergistic effect to coordinate tendon regeneration, such as mechanical loading to modulate the expression of a specific growth factor to optimize tissue healing through cell signaling.

### 6.1. Chemical Stimulation in Tendon Tissue Engineering

As tendons are composed of dense connective tissue, they typically have a limited intrinsic capability to heal. Various chemical factors have been studied to stimulate and accelerate the tendon healing process, including growth factors, cytokines, and cell adhesion molecules (CAMs). The goal of chemical stimulation in tendon repair is to mimic and enhance the natural signaling processes involved in tissue regeneration [23]. Notable growth factors that modulate cellular response during tendon repair include insulin-like growth factor-1 (IGF-1), transforming growth factor-beta (TGF-β), platelet-derived growth factor (PDGF), and vascular endothelial growth factor (VEGF), as outlined in Table 3 [24]. These various growth factors stimulate tendon tissue engineering by promoting cell proliferation, ECM synthesis, and tissue remodeling. 

During injury, tenocytes are exposed to an oxygen-deficient environment, which leads to a cascade of reactive oxygen species production and an acute spike in inflammatory markers as a response to the hypoxic milieu. The inflammatory process initiates the removal of necrotic debris as well as the proliferation of new tenocytes and type III collagen repair [21]. Four described signaling pathways leading to an inflammatory response in tendinopathy are NF-κB, NLRP3, p38/MAPK, and signal transducer and activator of transcription 3, as described in Table 4 [21]. NF-κB is arguably the most notable pro-inflammatory signaling pathway that drives the cascade of inflammatory cytokines, IL-1, IL-6, CCL2, and TNF-α. A persistent inflammatory response is cyclic, as these inflammatory markers are reactivated by NF-κB activity [21].

### 6.2. Mechanical Stimulation in Tendon Tissue Engineering

Mechanical stimulation in tendon tissue engineering requires the application of controlled and physiologically relevant mechanical forces to promote the development of functional and biomechanically sound tendon constructs closely mimicking the native tendon microenvironment. Tendon development and healing are activated by mechanical stretching, and mechanical stressors, such as tensile strain, shear force, and compression strength, can influence healing when applied to a tendon [23]. Tendon development and healing activated by mechanical stretching caused by tension is the main mechanical stimulus promoting the growth and development of tendons. Dynamic and static stretching are common forms of mechanical stimulation in the human body, and mechanical loading typically includes cyclic stretching and compression regimens. Mechanical stretching may have negative effects on engineered tendon tissue and may increase the diameter, elongate, or decrease the Young’s modulus of fabricated scaffolds [25].

The three main parameters that regulate dynamic tendon stretching in engineered tendon tissue are strain, frequency, and rest interval [26]. Mechanical stretching has the ability to induce tenogenic differentiation as well as promote osteogenesis, adipogenesis, and chondrogenesis based on the percentage of strain applied. Between 1 and 15% of strain has been shown to promote tendon differentiation. In in vivo models, the physiologic strain of dynamic stretching of a tendon is typically 4–8% [27]. High strain has the potential to cause premature cell differentiation and cell death, while low strain may result in disorganized cell differentiation. The ideal percentage of strain should be optimized for the specific condition desired in tendon healing.

The ideal frequency of mechanical stretching for tendon healing and engineering has been demonstrated to be 1 Hz, as it is the best condition to induce cell proliferation and tenogenic differentiation [25]. In contrast to higher frequency and shorter duration, the frequency rate of 1 cycle/minute and/or 0.5–1 h/day has been shown to promote type I collagen production and organization into collagen [26]. Cells will gradually adapt to a stimulus applied over time, therefore reducing the impact of the external mechanical stimulus. However, the introduction of a rest interval can restore the mechanical sensitivity of tenocytes. Mechanical stretching for greater than an hour per day may lead to limited gains due to the adaptation of the applied stimulus [26].

Mechanical stimulation leads to a cascade of tenocyte proliferation, migration, infiltration, and organization, as well as changes in the milieu of the ECM. Although mechanical stimulation is used to initiate the differentiation of stem cells into tenocytes, tendon fibers do not have specific markers for differentiation. Therefore, the expression of tendon-related transcription factors is important for tenocyte development and maturation, including Scleraxis, Mohawk, and Tenomodulin [26]. In addition to transcription factors, the milieu within the ECM can influence tendon healing and tenocyte maturation. The environment of the ECM changes in response to external stress and mechanical stimulation through the signaling of cell adhesion molecules, which attach to the cytoskeleton of tenocytes and react to mechanical stress by transmitting stimulus through the nuclear membrane and inducing gene expression [23].

## 7. Novel Therapeutic Approaches in Tendon Tissue Engineering

Throughout the past two decades, continued advancements have been noted in the fields of regenerative medicine, biological adjunctive therapy, and tendon tissue engineering for the treatment of tendon and ligament injuries [28], focusing on the paradigm of “cells, signals and scaffolds” [6]. Cell-based approaches include platelet-rich plasma (PRP), bone marrow aspirate concentrate (BMAC), and stem cell therapies, specifically ASCs.

While exosomes focus on the “signals” of the cells themselves as mediators to accelerate the local inflammatory cascade and regenerative response to tendinopathy, scaffolds provide a template for the formation of the ECM for cellular interactions. Recent developments in cell-based therapy, biological therapy, and gene therapy demonstrate promising early potential as adjuncts to native healing, mitigating the cascade of pro-inflammatory cytokines and scar-like tissue formation [29,30].

### 7.1. Cell-Based Therapy

Cell-based approaches have been implemented to treat tendinopathies with good effect, including agents such as PRP, autologous fibroblasts, tenocytes, and mesenchymal stem cells (MSCs) (or, as they have been referred to, “medicinal signaling cells”). Cell-based approaches demonstrate favorable effects, such as decreasing inflammation and modulating pain, therefore stimulating healing, particularly at the interface of the tendon to the bone. However, the most substantial limitation of these therapies and difficulty in comparison is due to the heterogeneity of reporting characteristics and preparation techniques [31].

Tenocyte migration, proliferation, and differentiation and ECM production are coordinated by cytokines and growth factor signaling. These growth factors are expressed by fibroblasts and inflammatory cells, which activate signaling and the transcription of genes that regulate tendon healing. By stimulating the proper cell signaling pathway, cytokines and growth factors can induce the stimulation or synthesis of collagen, ECM components, and/or angiogenesis in the final stage of remodeling for tendon healing [21,32]. The efficacy of cell-based therapy is based on the balance of modulating inflammation and cell remodeling directed by the sequential expression of growth factors and cytokines. These cells upregulate the response to tendon tissue healing through stimulating anabolic responses while restoring ECM homeostasis.

The use of exogenous cells to regulate healing includes the stimulation and signaling of preexisting cells through local injection of ASCs directly to the site of injury or the introduction of ASCs in the form of cytokines and growth factors to stimulate tendon regeneration [3]. Both embryonic and adult ASCs can be implemented in cell-based therapies. It can be difficult to control the differentiation of embryonic stem cells, which also have tumorigenic properties. Therefore, adult ASCs are the preferred source for tissue regeneration [33] and have been shown to be easier to isolate, culture, and expand [34]. Reprogramming differentiated cells with genetic modifications through the induction of pluripotent stem cells resembles embryonic stem cells but without the disorganized and uncontrolled differentiation and limited donor sources. Due to the plentiful source of MSCs in the human body, along with their multipotency lineage differentiation capability, MSCs are a powerful strategy for cell-based therapies in tendon tissue engineering.

MSCs have furthermore demonstrated favorable survival and proliferation characteristics when injected directly at the site of injury, which can be used as a target for vectors of gene therapy. Tenocytes and fibroblasts, which are autologous differentiated cells, can be injected directly at the site of defects or injury within tendon tissue defects or built into vesicles for implantation. However, the availability of donor sites for cell harvesting, lengthy times for culture expansion, and potential for donor site morbidity in autologous cell-based techniques are commonly cited limitations [24,35]. Additionally, optimizing cell delivery, achieving consistent differentiation, and standardizing regulatory processes also pose challenges.

In cell-mediated tendon repair, therapies are either injected directly at the site of tendon injury or implanted in a scaffolding or tissue graft construct to enable homogenous distribution and localization of repair cells at the site of tendon injury. There are a wide variety of options for cell-based therapy, including permanently differentiated cells (tenocytes, dermal fibroblasts, and/or muscle cells), undifferentiated progenitor cells (bone marrow, adipose-derived, embryonic, and/or tendon stem cells), and reprogrammed/engineered cells (induced pluripotent stem cells) [31].

There is no widely accepted method for optimal in vitro propagation of tenocytes. Accredited to the complexity of native tendon tissue and the lack of tendon-specific markers for healing, there is no universally accepted in vitro method to initiate the propagation of tenocytes, and a multifactorial approach is required to enable stable expansion and the use of cell-mediated therapy [31].

PRP is a cell-based therapy widely used to treat tendon injuries in the clinical setting with both anabolic and anti-inflammatory effects. PRP is advantageous due to its safety profile, widespread use, and simple preparation and administration methods that can be performed in a relatively noninvasive manner in the everyday clinical setting [36]. PRP secretes several growth factors to enhance tendon healing through the formation of a fibrin gel, providing a conductive bioscaffold landing for migrating cells for tendon healing [37]. Table 5 reviews recent clinical data, including level I and II studies that summarize the efficacy and utility of PRP injections in the treatment of tendinopathies. In general, there is a paucity of evidence to support PRP injection as a superior treatment method to conservative measures such as physical therapy, activity modification, and observation alone in the setting of tendinopathies. The most compelling evidence favoring PRP injection exists in the setting of rotator cuff tears, with some data that suggest lower retear rates and improved functional outcomes with respect to the percentage of rotator cuff tears. 

### 7.2. Exosome-Based Therapy

Cell-free therapies, particularly exosomes, are gaining increasing appreciation for their healing effects in directed tissue through paracrine-mediated actions [63,64]. Exosomes are small extracellular vesicles secreted by various cell types and play an integral role in intercellular communication through the transportation of proteins, lipids, and nucleic acids. These nanosized exosomes (30–140 nm) can cross cellular boundaries and facilitate intercellular signal delivery in various regenerative tendon processes [64]. These exosomes can also be stored for instant use in the operating room and produced in a cost-effective manner [30]. In comparison to cell-based therapies, exosomes do not carry the same logistical burden profile, such as harvest preparation, long culture expansion times, and donor site morbidity. Additional benefits of cell-free therapy include a lower immunogenetic response, a longer preserved shelf life, selective target tissue characteristics, and a better safety profile [32,65,66].

Exosome-based therapy has garnered increasing awareness in tendon tissue repair and regeneration, especially exosomes derived from MSCs. Cell-free therapy promotes tendon healing through its anti-inflammatory effects as well as stimulation of signaling pathways that modulate cellular responses and promote the proliferation and differentiation of tenocytes. Exosomes are found in almost all cell types, which is helpful in leading intercellular communication. In addition, they are highly stable, with a lipid bilayer that defends attached cargo against enzymatic degradation. 

ASC exosomes have gained interest due to the ease of isolation of adipose tissue through minimally invasive procedures and their high differentiation ability and immunomodulatory capacity [34]. The primary mechanisms by which ASCs promote tendon repair include the induction of angiogenesis, proliferation of tenocytes, assembly of anti-inflammatory cytokines, promotion of metabolic homeostasis, and formation of new collagen [66]. ASCs can be delivered directly to the tendon healing interface through injection into the injured tendon or delivered via fabricated scaffolds, and they contain a broad range of intracellular proteins that are engaged in extracellular signaling, acting on more than two hundred pathways. The role of ASCs in tenocyte proliferation, differentiation, and migration is illustrated in Figure 3 [34].

In addition to promoting primary proliferation and differentiation, tenocytes also stimulate the ECM microenvironment for healing. Exosomes have been shown to alter the expression of mRNA and subsequent protein transcription in a hypoxic injury environment, with upregulation of several matrix regenerative mediators [29]. Additionally, exosomes regulate the production of type I collagen and matrix metalloproteinases, which balance the remodeling environment of a tendon ECM [67]. Additionally, exosomes demonstrate both anti-inflammatory and anti-scar formation properties in tendon healing, most notably from their ability to facilitate macrophage responses with a dominant M2 phenotype pattern. M2 phenotype macrophages stimulate anti-inflammatory effects and rebuild the ECM [30]. 

Currently, limitations in quantification and standardization for large-scale production represent a conspicuous barrier to the widespread use of exosomes for the therapeutic treatment of tendon injuries [30,64]. Isolation and purification techniques to produce exosomes on a large scale are challenging, which is crucial due to the potential for immunogenicity. There is no current method of isolation that can isolate exosomes quickly and reliably on a large scale, and this fact represents a notable area for future innovation [33].

### 7.3. Scaffolds for Tendon Repair

Tissue-engineered scaffolding combined with injectable stem cell therapy can reform the mechanical structure and heal injured tendons. Tendon scaffolds require biocompatible and biodegradable materials to offer short-term stability for transplantation while also allowing for attachment to the host cell and integration into the host tissue. Fibril alignment must support the architecture of the scaffold and is important to transmit uniform force across the tendon [68]. Scaffolding materials must possess mechanical properties similar to those of the host tissue, promote rapid integration of the new tissue, and offer structural support to deliver nutrients for healing. Scaffolds must be made of biodegradable materials and degrade over time after being replaced by newly regenerated tissue [69]. Tendon scaffolds can be divided into biologic, synthetic, and composite polymers, with the advantages and disadvantages of each detailed in Table 6. A combination of these polymer scaffolds with cell-mediated therapy, such as stromal cells, tenocytes, and growth factors, has demonstrated significantly improved repaired tendon functional outcomes [70].

#### 7.3.1. Biological Scaffolds

Biological scaffolds are derived from human, porcine, bovine, and equine tissues. The most common material used as a biological scaffold includes collagen, the major component of tendons, or decellularized tissue, which retains the ECM structure (Figure 4) [71]. The major advantages of natural polymers include their biocompatible properties and native microenvironment with ECM and collagen. Collagen formed in combination with other polymers and MSCs provides the optimal microenvironment for the differentiation and maturation of stem cells into tenocytes within the scaffold matrix. In order to minimize host rejection while maintaining their complex collagenous structure and biomechanical properties, there is in-depth processing to remove cell components, fat, lipids, and endotoxins. Biological scaffolds are highly hydrophilic and have a low immune response. However, they have inferior mechanical properties to synthetic scaffolds [26,68].

#### 7.3.2. Synthetic Scaffolds

Synthetic scaffolds have similar biomechanical and physical properties as native tendons and include poly(lactide-*co*-glycolide) (PLGA), polyglycolide (PGA), and polylactide (PLA), which are reliably engineered with predictable biomechanical properties [24]. While synthetic scaffolds have stronger mechanical properties, they can ignite an immunological response during host integration and have an added risk of graft rejection, inflammatory reactions, and cell toxicity [24,72,73]. Synthetic scaffolds demonstrate more versatility; as a result, their biochemical and physical properties can be engineered with a high degree of precision [24]. There are two methods to optimize their precision: either optimizing the structure through scaffold material or reforming the chemistry of the native microenvironment (Figure 5) [72].

#### 7.3.3. Composite Scaffolds

Synthetic and biological polymer scaffolds can be combined to achieve optimal biological conditions and leverage the advantages of each construct. Examples include collagen–PLGA composite scaffolds as well as hybrid scaffolds, such as those combining natural collagen fibers with synthetic polymers [69]. Composite scaffolds are designed to account for early degradation, maximize structural support, and safeguard against an immunogenetic response in tendon tissue engineering. Composite polymer and ceramic scaffolds have the added advantage of osteoinductive properties, resistance to corrosion, and the ability to withstand higher compression forces. Furthermore, composite scaffolds exhibit synergistic effects by combining the bioactivity of biological materials with the tunability of synthetic materials [20].

The advantage of combining active biologics with scaffolds to promote tendon-to-bone healing is illustrated by localized recombinant human parathyroid hormone (rhPTH) delivery on a scaffold for targeted and controlled hormone release with a scaffold template to introduce fibrocartilage tissue ingrowth along the fibers [74,75,76]. There is less drug use, fewer off-target risks, and lower costs associated with this localized and targeted approach. Biocomposite rhPTH has demonstrated significant improvements in developing organized and denser collagen fiber formation with greater bone mineral density, increasing the ultimate load to failure in biomechanical testing [77]. The systemic effects of rhPTH coupled with bioengineered scaffolds to augment tendon repairs in an animal model demonstrate improved short-term integrity of a tendon repair and promote the formation of a more organized tendon-to-bone interface [74,75]. Interest in PTH as a targeted adjunct therapy to improve healing at the tendon-bone interface continues to develop, initiated with the introduction of biointegrative scaffolds to enhance rotator cuff repair healing [74,76]. PTH activates chondrogenesis and angiogenesis, in addition to preventing fatty infiltration within the rotator cuff. In an effort to limit systemic toxicities, rhPTH is a potent growth factor that can be delivered effectively through standard IV administration [74].

An Individualized approach towards selecting the optimal scaffold composite should be determined based on the relative advantages and disadvantages. The ideal scaffold material should immolate the mechanical and biological properties of the host tendon tissue to restore the native microcellular environment for cell adhesion, growth, differentiation, and maturation. 

## 8. Future Directions with Novel Advancements

Despite the advances in tendon engineering, researchers have been unable to completely regenerate native tendon tissue due to its complex composition and structure. The field of tendon engineering and regeneration is rapidly evolving, with promising future advances being explored, such as magnetic stimulation and gene therapy. 

Magnetic stimulation has great potential to improve performance in tissue engineering through the force of the magnetic field itself as well as the indirect mechanical action produced by a magnetic field through magnetic particles [78]. External magnetic fields offer a non-invasive, targeted approach to modulate cellular response through the induction of controlled mechanical stimulation, which further promotes a cellular response pathway for tissue regeneration. Mechanosensitive nanoparticles can be incorporated into engineered scaffolds to allow for dynamic modulation of scaffolds in vivo, allowing for real-time adaptation to the changing needs for tendon healing. 

A low-frequency magnetic field can be used to adjust the pro-inflammatory response in tendon healing. Additionally, a low-frequency static magnetic field has been shown to increase the expression of tendon-related genes and proteins through the alteration of intracellular calcium ion concentration and the activation of oxygen release [79]. The ability to trigger a tendon-related transcription response and promote differentiation of tenocytes without the need for scaffold support can avoid the potential side effects of mechanical stimulation, which include increased pore size, elongation, and decreased elasticity [80]. Additionally, magnetic force stimulation can effectively apply mechanical stimulation and regulate the inflammatory response, promoting enhanced biological performance. Furthermore, magnetic force simulation can be adjusted remotely by changing the magnetic field to precisely deliver mechanical stimulation to tissue cells [23].

Gene therapy is also being explored as a mechanism to directly introduce the genes associated with tenogenic differentiation directly into an injured tendon. This process involves the local production of therapeutic proteins to enhance tendon healing by introducing genetic material encoding specific growth factors or signaling molecules into the injured tissues [64]. By encoding tenogenic transcription factors or signaling molecules, regeneration of tendon tissue can be achieved through the differentiation of cells towards a tenocyte phenotype. 

The development of advanced biomaterials, including “smart” scaffolds and exosomes, as well as bioactive coatings, is being explored as future advancements in tendon tissue engineering [81]. By developing scaffolds and exosomes with responsive properties, these constructs can adapt their mechanical properties, release bioactive molecules, and respond to cues in the ECM to imitate the native dynamic microenvironment of healing tendon tissue. The ability to reprogram a “smart” scaffold or exosome based on the pathological milieu and to customize exosome content for the delivery of therapeutic molecules is still in its early stages in regard to tissue tendon engineering [72,82]. However, the controlled manipulation of cell-mediated therapy and tissue scaffolds by modifying the cargo and surface modifications for targeting, delivery, and diagnostic purposes has enormous translational opportunities in the management of tendon injury. Ongoing research and technological advancements are continuing to contribute to the translation of tendon tissue engineering cell-mediated and cell-free therapy into clinically viable solutions for the treatment of tendon injuries.

## 9. Conclusions

NBMSKIs represent one of the most immediate threats to medical readiness and deployability among servicemembers, despite the recent decrease in high-tempo combat warfare. Tendon injuries are one of the more common overuse injuries, leading to a significant source of pain and dysfunction and posing a significant threat to servicemembers’ operational readiness. This illustrates the need for continued efforts to optimize medical treatment to accelerate recovery and return to unrestricted duty. Funding and stakeholder support for coordinated injury surveillance efforts and prevention programs are needed to maintain the lethality of a fighting force. In addition to being a major cause of medical disability, NBMSKIs can lead to a high percentage of medical discharges and pose a significant financial burden to the MHS from service-connected disability costs.

Understanding the complex structure, healing properties, and risk factors associated with tendinopathy and tendon injuries is essential for the development of optimal prevention and treatment tactics. Exosome-based therapy, cell-based therapy, and scaffolds may have synergistic effects and capitalize on the advantages of each method if employed together for tendon healing. Ongoing research aims to translate these tissue tendon engineering strategies into effective and durable clinical treatments.

## Figures and Tables

**Figure 1 bioengineering-11-00158-f001:**
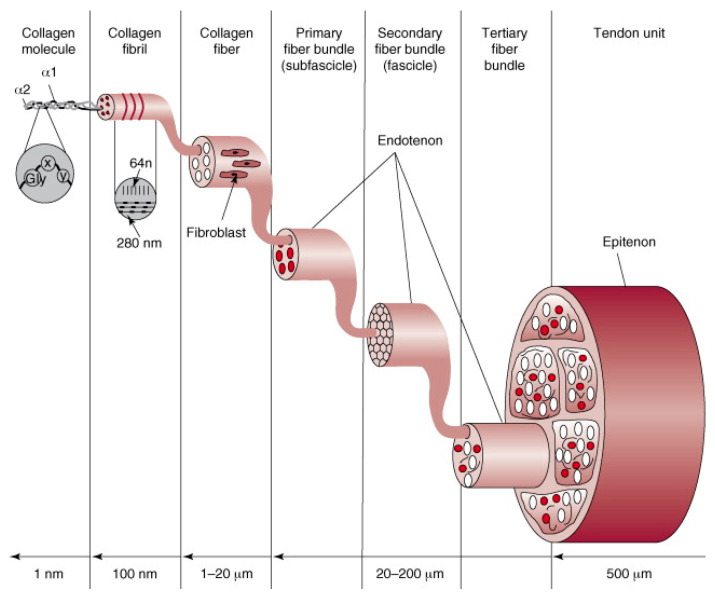
The basic structure of the tendon unit. Reproduced with permission from Wang JHC, J Biomech; published by Elsevier, 2006 [17].

**Figure 2 bioengineering-11-00158-f002:**
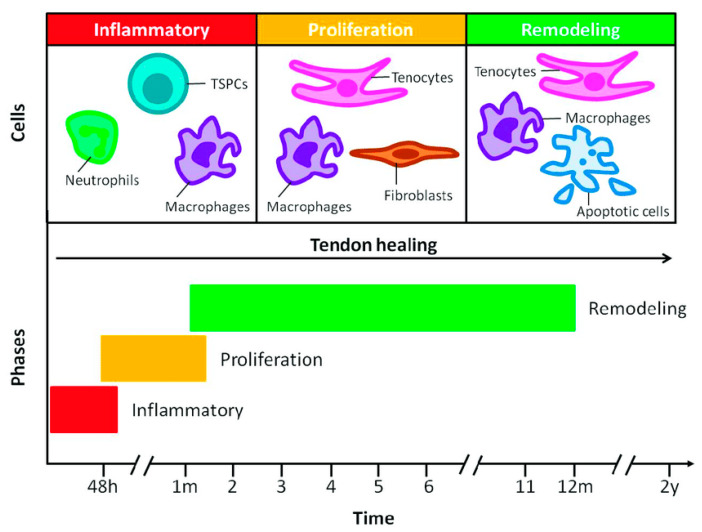
Phases of tendon healing. Reproduced with permission from Vasiliadis AV, Katakalos K, J Funct Biomater; published by MDPI (Basel, Switzerland), 2020. http://creativecommons.org/licenses/by/4.0/ (accessed on 18 December 2023) [20].

**Figure 3 bioengineering-11-00158-f003:**
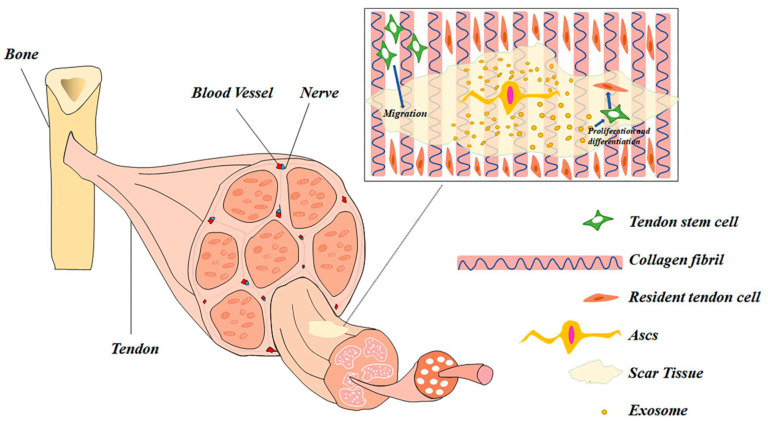
The role of adipose-derived stem cells (ASCs) in tenocyte maturation. Reproduced with permission from Lyu K, Liu T, Chen Y, et al., Eur J Med Res; published by Springer Nature, 2022. http://creativecommons.org/licenses/by/4.0/ (accessed on 14 December 2023) [34].

**Figure 4 bioengineering-11-00158-f004:**
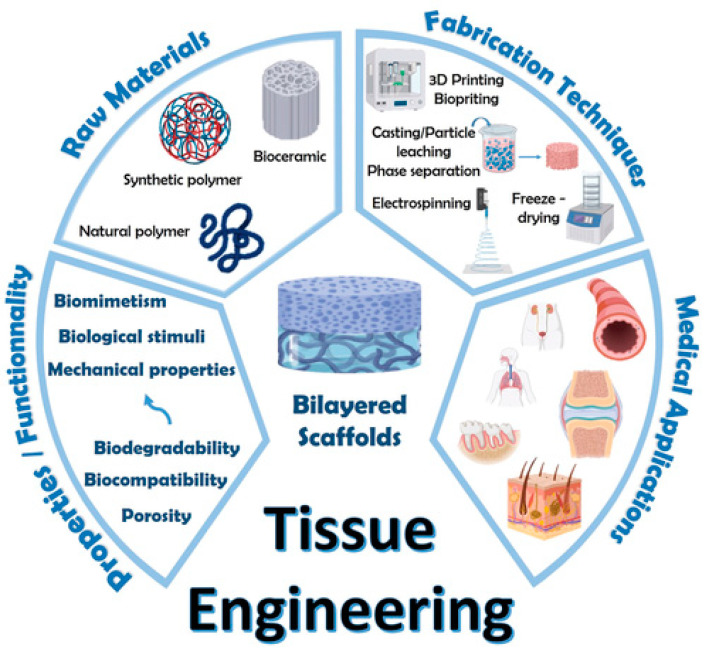
Components of biological scaffolds and fabrication for tendon tissue engineering. Reproduced with permission from Bertsch C, Maréchal H, Gribova V, et al., Advanced Healthcare Materials; published by John Wiley and Sons, 2023. http://creativecommons.org/licenses/by/4.0/ (accessed on 14 December 2023) [71].

**Figure 5 bioengineering-11-00158-f005:**
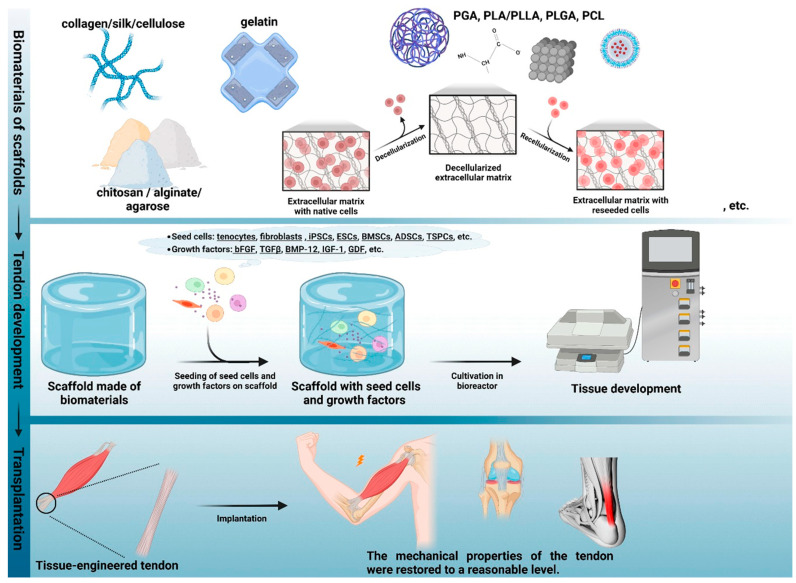
Interaction between scaffold biomaterials and microenvironment for tendon tissue engineering. Reproduced with permission from Huang L, Chen L, Chen H, et al., Biomimetics; published by MDPI (Basel, Switzerland), 2023. http://creativecommons.org/licenses/by/4.0/ (accessed on 14 December 2023) [72].

**Table 1 bioengineering-11-00158-t001:** The seven principles of load management for injury prevention.

Principle of Load Management	Components for Injury Prevention
Establish a moderate chronic load	Routine monitoring (i.e., accelerometers) Individualizing training programs
Lessen abrupt weekly changes	Balancing resources and demands Gradually increasing physical fitness Standardizing pre-deployment fitness assessments Offering training programs geared for physical fitness assessments
Avoid the safety workload ceiling	Appropriately planning variation in training intensity and/or volume Defining contemporary fitness standards
Enforce a standard minimum training requirement	Providing training protocols Self-responsibility for compliance
Avoid inconsistent “boom-bust” workloads	Temporarily adjusting specific activity restrictions in setting of injury
Establish consistent training schedules proportionate to workload demands	Conscious planning of training loads Variability in exercises, standards, and distances based on demands of specific occupations Defining demands and standards of each military occupational task
Monitor servicemembers throughout the maintenance phase	Routinely measuring physical and duty/occupation-related competency Timely identifying injuries and rehabilitation needs Continual monitoring of training loads

**Table 3 bioengineering-11-00158-t003:** Growth factors stimulating tendon repair.

Growth Factor	Abbreviation	Purpose
Insulin-like growth factor-1	IGF-1	Tendon fibrinogenesis, stimulates cell proliferation and matrix synthesis
Transforming growth factor-beta	TGF-β	Tendon remodeling, promotes production of collagen and ECM components
Platelet-derived growth factor	PDGF	Recruitment and activation, stimulates proliferation and synthesis of collagen
Vascular endothelial growth factor	VEGF	Angiogenesis, facilitates adequate blood supply to deliver nutrients and oxygen for healing
Interleukins	IL-1, IL-6	Pro-inflammatory mediation, controls inflammatory response to clear necrotic tissue and initiates repair
Connective tissue growth factor	CTGF	Tissue remodeling and scar formation, contributes to synthesis of ECM proteins and collagen

ECM = extracellular matrix.

**Table 4 bioengineering-11-00158-t004:** Inflammatory signaling pathways in response to tendon injury.

Growth Factor	Abbreviation	Purpose
Nuclear factor-kappa B	NF-κB	Dominant pro-inflammatory pathway involved in all cycles of tendon healing: inflammation, cell proliferation, angiogenesis, and scar formation Incites production of both pro-inflammatory cytokines and chemokines Contributes to sustained inflammation and tissue damage
NOD-like receptor family, pyrin domain-containing 3	NLRP3	Inflammatory vesicle complex that promotes cyto-kine expression and disarray of ECM components, promoting the maturation and release of IL-1*β* Activation leads to release of pro-inflammatory cytokine
Mitogen-activated protein kinase	p38/MAPK	Mechanical stress-activated pathway with consecutive phosphorylation events Associated with the production of inflammatory mediators and regulation of MMPs involved in tissue remodeling
Signal transducer and activator of transcription 3	STAT3	Dual regulatory role, linked to the regulation of inflammatory responses and the promotion of cell survival

ECM = extracellular matrix; MMPs = matrix metalloproteinases.

**Table 5 bioengineering-11-00158-t005:** Recent studies investigating clinical outcomes of PRP in the setting of tendinopathy.

Reference	Year	Sample Size	Study Design	Study Group	Study Conclusion
Droppelmann [38]	2022	318	Metanalysis (8 RCTs)	Achilles tendinopathy/patellar tendinopathy	No difference in outcome score (VISA-A) between Achilles and patellar tendinitis groups No difference based on type of regenerative therapy (stem cells or PRP) or injection site
Madhi [39]	2020	230	Systematic review	Achilles tendinopathy	Significant improvement in baseline VISA-A score from 41 to 70 post-treatment (*p* = 0.018)
Scott [40]	2019	57	RCT	Patellar tendinopathy	No difference in change of VISA-P score, pain, or global rating across treatment groups (LR-PRP, LP-PRP, or no treatment) at 12 weeks or up to 1 year after treatment combined with exercise program
Desouza [41]	2023	NR	Metanalysis (5 RCTs)	Achilles tendinopathy	No difference in VISA-A score between PRP and placebo groups at 12 weeks or up to 1 year after treatment PRP with better efficacy than placebo at 6 weeks after treatment
Alsousou [42]/Keene [43]	2019/2022	230	RCT	Achilles tendon rupture	No difference in function between PRP and placebo group or adverse event rates PRP did not improve patient-reported functional outcomes or quality of life 2 years after acute Achilles tendon rupture compared with placebo
Wang [36]	2021	363	Systematic review (5 RCTs)	Achilles tendon rupture	PRP had positive effects (no significant difference) on ankle dorsiflexion, strength, and calf circumference compared with controls
Vithran [44]	2023	526	Metanalysis (8 RCTs)	Achilles tendinopathy	No difference in function between PRP and placebo group or adverse event rates PRP did not improve PROs or quality of life 2 years after acute AT rupture compared with placebo
Kearney [45]	2021	240	RCT	Achilles tendinopathy	PRP injection compared to placebo did not improve Achilles tendon dysfunction No difference in outcome score (VISA-A) at 6 months between PRP and placebo
Boesen [46]	2020	40	RCT	Achilles tendon rupture	No difference in ATRS score between PRP and placebo groups at any time point No differences in functional outcomes at any time point between the study groups
Rodas [35]	2021	20	RCT	Patellar tendinopathy	No difference in outcome score (VISA-A) between BM-MSC and LP-PRP groups BM-MSC treatment resulted in improvement in tendon structure compared with LP-PRP at 6 months
Barman [47]	2022	123	Metanalysis (5 RCTs)	Patellar tendinopathy	No difference in QoL in short- or long-term follow-up based on treatment group No difference in pain relief between PRP and placebo groups
Nuhmani [48]	2022	338	Metanalysis (9 RCTs)	Patellar tendinopathy	Similar outcome scores (VISA-P) between PRP and other non-injection therapies (eccentric training, ESWT, and arthroscopy)
Chen [49]	2019	430	Metanalysis	Patellar tendinopathy	LR-PRP has the largest functional improvement and reduction in pain reduction compared with other treatment options (CSI, ESWT, US, AWB, dry needling) Treatment outcomes may be biased by intransitivity and should not be overestimated
Andriolo [50]	2018	2530	Metanalysis	Patellar tendinopathy	Eccentric exercise demonstrates best results in the short term (<6 months) Recurrent injections of PRP obtained best functional results followed by ESWT and eccentric exercise at long-term follow-up ≥6 months
Dai [51]	2023	576	Metanalysis (13 RCTs)	Lateral epicondylitis, rotator cuff tendinopathy, patellar tendinopathy	No significant difference in pain relief or functional improvement between PRP and placebo from 4 weeks to 6 months after treatment No difference in pain relief or functional outcomes based on type of tendinopathy, treatment regimen, leukocyte concentrations, or cointerventions
Wong [52]	2022	1520	Systematic review (20 RCTs)	Lateral epicondylitis	Limited robust evidence to recommend PRP therapy over other treatments (physiotherapy, CSI, AWB, and surgical interventions) Potential confounder is heterogeneity of PRP formulations
Watts [53]	2018	81	RCT	Lateral epicondylitis	PRP and surgery produced same functional outcomes 70% of patients treated with PRP avoided surgical treatment 16% of patients transferred from PRP cohort to surgery by 12 months
Kandil [54]	2022	120	RCT	Lateral epicondylitis	At 3-month follow-up, improvement in PRTEE and qDASH scores were 88% and 70% in the PRP group (versus 21% and 14% in the control group) At final follow-up, PRP led to excellent outcomes in 85% of patients and good outcomes in 15% of patients (versus 8% and 32% in the control group) Overall, there was a 95% patient satisfaction rate in the PRP group compared to 25% in the control group
Simental-Mendía [55]	2020	276	Metanalysis (5 RCTs)	Lateral epicondylitis	No changes were noted for pain between PRP and placebo PRP and placebo had equivalent outcomes in terms of pain and function
Muthu [56]	2022	2040	Metanalysis (25 RCTs)	Lateral epicondylitis	LR-PRP offers notable pain improvement compared to control without similar improvement in functional outcomes LR-PRP was equivalent to the control in terms of outcome scores (DASH and PRETEE scores)
Oudelaar [57]	2021	80	RCT	Rotator cuff tears and tendonitis	NACD + PRP led to inferior clinical outcomes at 6-week follow-up but superior clinical outcomes at 6-month follow-up compared to NACD + CSI Clinical results were similar between groups at 1- and 2-year follow-up
Prodromos [58]	2021	71	Prospective cohort study	Rotator cuff tears and tendonitis	PRP injection is safe and effective after failed conservative therapies (activity modification and physical therapy) without a decline in outcomes at 2-year follow-up Patients with >50% partial tear showed the best overall improvement, while the tendinitis group had the poorest outcome scores
Chen [59]	2019	1116	Metanalysis (18 RCTs)	Rotator cuff tears and tendonitis	Rates of retear were lower in patients who received PRP at long-term follow-up The PRP-treatment group demonstrated noted improvement in multiple functional outcomes; however, none reached their respective MCIDs
Hurley [60]	2019	1147	Metanalysis (18 RCTs)	Rotator cuff tears (undergoing arthroscopic repair)	PRP resulted in lower rates of incomplete tendon healing for all tears combined compared to the control group (17% vs. 30%, respectively; *p* < 0.05) PRP led to improved Constant and pain scores at 30 days postoperatively and final follow-up compared to control PRP did not improve tendon healing or Constant scores compared to the control, with longer operative times compared to the control
Xiang [61]	2021	629	Metanalysis (9 RCTs)	Rotator cuff tears and tendonitis	Short-term effects of PRP were significant in terms of pain relief, Constant–Murley score, and SPDI No long-term effect was observed on pain and function, except Constant–Murley score Differences in pain relief were significant in PRP groups treated with double centrifugation, single injection, and post-injection rehabilitation
Godek [62]	2022	90	RCT	Rotator cuff tears, partial thickness	Collagen and PRP combined therapies demonstrated comparable outcomes with monotherapy in either collagen or PRP alone

ATRS = Achilles tendon total rupture score; AWB = autologous whole blood; BM-MSC = bone marrow-derived mesenchymal stem cell; CSI = corticosteroid injection; ESWT = extracorporeal shockwave therapy; LR-PRP = leukocyte-rich PRP; LP-PRP = leukocyte-poor PRP; MCIDs = minimal clinically important differences; NACD = needle aspiration of calcific deposits; PRTEE = patient-rated tennis elbow evaluation; PROs = patient-reported outcomes; PRF = platelet-rich fibrin; PRP = platelet-rich plasma; QoL = quality of life; qDASH: quick disabilities of arm, shoulder, and hand; RCT = randomized control trial; SPDI = Shoulder Pain and Disability Index; US = ultrasound; VISA-A = Victorian Institute of Sport Assessment, Achilles tendon; VISA-P = Victorian Institute of Sport Assessment, patellar tendon.

**Table 6 bioengineering-11-00158-t006:** Advantages and disadvantages of scaffolds for tendon repair.

Type	Tissue Source	Advantages	Disadvantages
Biological	Decellularized matrix Collagen tissue	Biocompatible, bioactive Low immunogenicity Biodegradable	Limited standardization Inferior mechanical properties Batch-to-batch variability
Synthetic	Polyesters Polyurethanes	Superior mechanical properties Customization Low batch-to-batch variability Consistent reproducibility	High immunogenicity Non-bioactive Complex fabrication Difficulty host integration
Composite	Collagen-PLGA Hybrid tissue	Synergistic effects Customization Enhanced biocompatibility Improved host integration	Increasing complexity Production costs Lack of standardization

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
