# Peer review of "Treatment of Tendon Injuries in the Servicemember Population across the Spectrum of Pathology: From Exosomes to Bioinductive Scaffolds"

_bioengineering, 2024, doi:10.3390/bioengineering11020158_

Round 1

Reviewer 1 Report

Comments and Suggestions for Authors

This review article entitled “Treatment of Tendon Injuries Across the Spectrum of Pathology: From Exosomes to Bioinductive Scaffolds” by DeFoor describes structure of tendon and cell biology of tendon injury and repair process, and advances in treatment of tendon injuries. Unfortunately, this review seems to lack originality: All the figures are adopted from assortment of other review articles. Whether or not these authors are qualified to summarize the field is questionable because a large proportion of references are review articles. Other concerns are listed below. Overall, this article is not recommended for publication.

Major Points:

L329:   This doesn't match with the study conclusions listed in Table5. Vast majority of references concluded "no difference". There are only small proportion of studies concluded "significant improvement".

First 3 pages of the article were devoted to emphasizing how common tendon injury is about military service members, and what are the management principles applied. This article is more appropriate for the internal magazine for military service members rather than a peer review scientific journals such as Bioengineering.

Table 2,3&4: Each table cites one review article. This is not appropriate for a review article. For each function of each growth factor etc. should cite original articles.

Minor Points:

L108:   Although the subtitle mentions “warfighter lethality”, it is not clear how the content relates to lethality.

L142:   There should be an illustration of tendon structure here to introduce the main content of the review.

Table 2:  For the effects, citations should be added in addition to the main text.

Author Response

Reviewer #1:

This review article entitled “Treatment of Tendon Injuries Across the Spectrum of Pathology: From Exosomes to Bioinductive Scaffolds” by DeFoor describes structure of tendon and cell biology of tendon injury and repair process, and advances in treatment of tendon injuries. Unfortunately, this review seems to lack originality: All the figures are adopted from assortment of other review articles. Whether or not these authors are qualified to summarize the field is questionable because a large proportion of references are review articles. Other concerns are listed below. Overall, this article is not recommended for publication.

The intended purpose of this invited review article was to provide a comprehensive overview of the current concepts in tendon tissue engineering in the context of the tactical athlete and the implications for readiness and maintaining lethality of the servicemember population. The purpose of this article was not to provide original research or data of our own, but to illustrate the most up-to-date concepts and future focus. Thank you for your feedback below which has been addressed to improve our manuscript for publication.

Major Points:

L329:   This doesn't match with the study conclusions listed in Table5. Vast majority of references concluded "no difference". There are only small proportion of studies concluded "significant improvement".

This sentence has been revised (lines 338-341). We agree that the vast majority of references conclude no difference in the use of PRP. However, the point we are trying to make is that out of all of the locations for major tendon injury, PRP use is most supported in the setting of partial rotator cuff tear, based on the available literature.

First 3 pages of the article were devoted to emphasizing how common tendon injury is about military service members, and what are the management principles applied. This article is more appropriate for the internal magazine for military service members rather than a peer review scientific journals such as Bioengineering.

As discussed above, the intended purpose of this review article was to provide a comprehensive overview of the current concepts in tendon tissue engineering in the context of the tactical athlete and the implications for readiness and maintaining lethality of the servicemember population. In the setting of this special issue regarding Operational Medicine Applications of Bioengineering, including context of the incidence, prevalence and impact of tendon injury in the servicemember population sets the stage for the importance of the bioengineering principles of tissue tendon engineering that can be applied to prevent, detect and treat arguably the single most common musculoskeletal injury faced in the austere environment. Respectfully, therefore, no further changes were made.

Table 2,3&4: Each table cites one review article. This is not appropriate for a review article. For each function of each growth factor etc. should cite original articles.

This review was meant to provide a comprehensive overview and basic introduction. Each of these topics has the potential to be a multi-paged review article on their own. We purposefully did not take a deeper dive into the original, basic science principles of the pathways, cytokines, growth factors, etc. presented in these tables, as that was outside the scope of this work.

Minor Points:

L108:   Although the subtitle mentions “warfighter lethality”, it is not clear how the content relates to lethality.

This section is designed to outline the impact of tendon injuries on the ability of servicemembers to be prepared and ready for operational duties in combat and austere environments. A major key to maintaining warfighter lethality is readiness and therefore prevention is a key aspect. No additional changes were made to this text.

L142:   There should be an illustration of tendon structure here to introduce the main content of the review.

Thank you for this feedback to assist with the clarity and flow of our review article. A figure has been added to illustrate the basic tendon structure (Figure 1), lines 159-160.

Table 2:  For the effects, citations should be added in addition to the main text.

Citation was added for Table 2 in addition to the main text.

Reviewer 2 Report

Comments and Suggestions for Authors

The author described emerging technologies for improving the treatment of tendon injuries with a special focus on exosome-based therapies, cell-based therapies, and bioinductive scaffolds. 

The review is comprehensive, balanced, well presented and well written

The reviewer recommends acceptance of this manuscript for publication

Author Response

Reviewer #2:

The author described emerging technologies for improving the treatment of tendon injuries with a special focus on exosome-based therapies, cell-based therapies, and bioinductive scaffolds. 

The review is comprehensive, balanced, well presented and well written.

The reviewer recommends acceptance of this manuscript for publication.

Thank you for your feedback and appreciation for the scope and purpose of our review. We look forward to adding to the knowledge base of the readership in this special edition on Operational Medicine Applications of Bioengineering.

Reviewer 3 Report

Comments and Suggestions for Authors

1.       The use of acronyms is inconsistent throughout the document. Please, revise.

2.       Introduction Section: lines 26 through 72 provide extremely thorough background information on an issue that is much broader than the one that should be the focus of the paper, based on its title. In other words, there is a lot of statistics shown in those 47 lines, which do not directly refer to tendon injuries. This can be highly confusing, because musculoskeletal injuries include damage to skeletal muscles, bones, tendons, joints, ligaments, etc. Please, revise the introduction to make the background information more specific to tendon injuries.

3.       Lines 73 through 107: although providing background information is crucial to outline the research problem, the excessive number of statistics that are presented here significantly impedes reading comprehension. Please, try to be more concise regarding the statistics, showing only those that are most relevant to the topic. Otherwise, the reader can easily get lost in a maze of numbers.  

4.       What is the relevance of Table 1 (and lines 129-140) to the topic “Treatment of Tendon Injuries Across the Spectrum of Pathology”? It honestly seems out of place. Please, revise.

5.       Magnetic stimulation therapy is unexpectedly presented in the Future Directions section, which significantly and negatively impacts coherence and flow. There is no connection between the first half of this section and the information presented in the previous paragraphs. Please, revise.

6.       The previous comment also applies for the “gene therapy” portion of the Future Directions section.

7.       What are the future directions regarding cell-based, exosome-based, and scaffold-based therapies? This should be included in the revised version of the manuscript. The authors failed to discuss the future perspectives of all the therapy approaches that were thoroughly presented in previous sections.  

8.       The Conclusions section lacks depth and connection to the Future Directions section. Please, revise.  

Author Response

Reviewer#3:

  1. The use of acronyms is inconsistent throughout the document. Please, revise.

Thank you for this feedback. All acronyms have been updated and are used consistently throughout the manuscript, most notably NBMSKI and ECM.

  1. Introduction Section: lines 26 through 72 provide extremely thorough background information on an issue that is much broader than the one that should be the focus of the paper, based on its title. In other words, there is a lot of statistics shown in those 47 lines, which do not directly refer to tendon injuries. This can be highly confusing, because musculoskeletal injuries include damage to skeletal muscles, bones, tendons, joints, ligaments, etc. Please, revise the introduction to make the background information more specific to tendon injuries.

The intended purpose of this review article was to provide a comprehensive overview of the current concepts in tendon tissue engineering in the context of the tactical athlete and the implications for readiness and maintaining lethality of the servicemember population. In the setting of this special issue regarding Operational Medicine Applications of Bioengineering, the context of the incidence, prevalence and impact of tendon injury in the servicemember population sets the stage for the bioengineering principles of tissue tendon engineering that can be applied to prevent, detect and treat arguably the single most common musculoskeletal injury in the austere environment. Respectfully, therefore, no further changes were made.

  1. Lines 73 through 107: although providing background information is crucial to outline the research problem, the excessive number of statistics that are presented here significantly impedes reading comprehension. Please, try to be more concise regarding the statistics, showing only those that are most relevant to the topic. Otherwise, the reader can easily get lost in a maze of numbers.  

As discussed above, the context of the incidence, prevalence and impact of tendon injury in the servicemember population sets the stage for the bioengineering principles of tissue tendon engineering that can be applied to prevent, detect and treat arguably the single most common musculoskeletal injury in the austere environment. The background information and statistics as related to the burden of disease and impact on operational readiness provides clinic context surrounding tendon tissue engineering in the military population. Respectfully, therefore, no further changes were made.

  1. What is the relevance of Table 1 (and lines 129-140) to the topic “Treatment of Tendon InjuriesAcross the Spectrum of Pathology”? It honestly seems out of place. Please, revise.

This special issue specifically focuses on the detection and prevention of this commonly diagnosed warfighter injury, therefore, it is important to highlight clinical relevance on preventive strategies and detecting training overload which is very applicable to overuse tendon injuries in the tactical athlete.  The title of the subsection was revised for clarity to better categorize this information, line 109.

  1. Magnetic stimulation therapy is unexpectedly presented in the Future Directions section, which significantly and negatively impacts coherence and flow. There is no connection between the first half of this section and the information presented in the previous paragraphs. Please, revise.

 The purpose of this section was not to expand upon the future direction of the previously discussed material, but rather to stand alone and briefly introduce “novel” and new “cutting edge” targets for tendon tissue engineering. Therefore, the name of this section was revised for clarity to better reflect the intent, “Future Directions with Novel Advancements”, line 482.

  1. The previous comment also applies for the “gene therapy” portion of the Future Directions section.

As discussed above the purpose of this section was not to expand upon the future direction of the previously discussed material, but rather to stand alone and briefly introduce “novel” and new “cutting edge” targets for tendon tissue engineering. Therefore, the name of this section was revised for clarity to better reflect the intent, “Future Directions with Novel Advancements”, line 482.

  1. What are the future directions regarding cell-based, exosome-based, and scaffold-based therapies? This should be included in the revised version of the manuscript. The authors failed to discuss the future perspectives of all the therapy approaches that were thoroughly presented in previous sections.  

 Due to the length of this comprehensive review, we felt that discussion of the future directions of each of these therapies was too exhaustive to include at the end of the manuscript, and instead was best addressed briefly throughout the manuscript in each subsection. Therefore, no additional changes were made.

  1. The Conclusions section lacks depth and connection to the Future Directions section. Please, revise.  

As discussed above, the name of the Future Direction section was revised to better reflect the intent, “Future Directions with Novel Advancements”, lines 482. Due to the length of this comprehensive review, we purposely aimed for brevity in the Conclusion section. Therefore, no additional changes were made.

Round 2

Reviewer 1 Report

Comments and Suggestions for Authors

The authors addressed all the points raised by this reviewer.

Author Response

Thank you for the thoughtful review and feedback provided to improve our manuscript

Reviewer 3 Report

Comments and Suggestions for Authors

1.       Introduction Section: lines 26 through 72 provide extremely thorough background information on an issue that is much broader than the one that should be the focus of the paper, based on its title. In other words, there is a lot of statistics shown in those 47 lines, which do not directly refer to tendon injuries. This can be highly confusing, because musculoskeletal injuries include damage to skeletal muscles, bones, tendons, joints, ligaments, etc. Please, revise the introduction to make the background information more specific to tendon injuries.

2.       Lines 73 through 107: although providing background information is crucial to outline the research problem, the excessive number of statistics that are presented here significantly impedes reading comprehension. Please, try to be more concise regarding the statistics, showing only those that are most relevant to the topic. Otherwise, the reader can easily get lost in a maze of numbers. 

Author Response

Introduction Section: lines 26 through 72 provide extremely thorough background information on an issue that is much broader than the one that should be the focus of the paper, based on its title. In other words, there is a lot of statistics shown in those 47 lines, which do not directly refer to tendon injuries. This can be highly confusing, because musculoskeletal injuries include damage to skeletal muscles, bones, tendons, joints, ligaments, etc. Please, revise the introduction to make the background information more specific to tendon injuries.

This section has been revised and condensed (lines 63-74) to include only background information that demonstrates the impact on medical readiness, depolyability and the associated financial burned on the military healthcare system. The language has also been revised to clarify that tendon injuries are the leading cause of non-battle musculoskeletal injuries.

Lines 73 through 107: although providing background information is crucial to outline the research problem, the excessive number of statistics that are presented here significantly impedes reading comprehension. Please, try to be more concise regarding the statistics, showing only those that are most relevant to the topic. Otherwise, the reader can easily get lost in a maze of numbers.  

This section has been revised and condensed (lines 83-99) to include only the major statistics related to the prevalence of major tendon ruptures, focusing on Achilles, patellar and pectoralis major tendons. 

Round 3

Reviewer 3 Report

Comments and Suggestions for Authors

The authors have reasonably addressed the issues that were raised. Thank you.